# Septoria Leaf Blotch and Reduced Nitrogen Availability Alter WRKY Transcription Factor Expression in a Codependent Manner

**DOI:** 10.3390/ijms21114165

**Published:** 2020-06-11

**Authors:** Alistair A. Poll, Jack Lee, Roy A. Sanderson, Ed Byrne, John A. Gatehouse, Ari Sadanandom, Angharad M. R. Gatehouse, Martin G. Edwards

**Affiliations:** 1School of Natural and Environmental Sciences, Newcastle University, Newcastle upon Tyne NE1 7RU, UK; alistair.poll@newcastle.ac.uk (A.A.P.); jackalexanderlee@hotmail.com (J.L.); roy.sanderson@ncl.ac.uk (R.A.S.); a.m.r.gatehouse@newcastle.ac.uk (A.M.R.G.); 2Department of Biosciences, Durham University, Durham DH1 3LE, UK; j.a.gatehouse@dur.ac.uk (J.A.G.); ari.sadanandom@dur.ac.uk (A.S.); 3KWS UK Ltd., 56 Church St, Thriplow, Royston SG8 7RE, UK; ed.byrne@kws-uk.com

**Keywords:** Septoria leaf blotch, stress, transcription factor, WRKY, wheat, *Zymoseptoria tritici*

## Abstract

A major cause of yield loss in wheat worldwide is the fungal pathogen *Zymoseptoria tritici*, a hemibiotrophic fungus which causes Septoria leaf blotch, the most destructive wheat disease in Europe. Resistance in commercial wheat varieties is poor, however, a link between reduced nitrogen availability and increased Septoria tolerance has been observed. We have shown that Septoria load is not affected by nitrogen, whilst the fungus is in its first, symptomless stage of growth. This suggests that a link between nitrogen and Septoria is only present during the necrotrophic phase of Septoria infection. Quantitative real-time PCR data demonstrated that WRKYs, a superfamily of plant-specific transcription factors, are differentially expressed in response to both reduced nitrogen and Septoria. *WRKY39* was downregulated over 30-fold in response to necrotrophic stage Septoria, whilst changes in the expression of *WRKY68a* during the late biotrophic phase were dependent on the concentration of nitrogen under which wheat is grown. WRKY68a may therefore mediate a link between nitrogen and Septoria. The potential remains to identify key regulators in the link between nitrogen and Septoria, and as such, elucidate molecular markers for wheat breeding, or targets for molecular-based breeding approaches.

## 1. Introduction

With over 750 million tonnes produced annually, wheat is the third most produced crop globally [1]. As such, wheat will be at the heart of the challenge to ensure the future of global food security. Estimates of the extent to which food production must increase in order to meet these growing demands vary, but it is thought that by 2050, global agricultural output must approximately double that of the levels recorded in 2005, if we are to meet the demands of an increasing population [2,3]. Therefore, sustainable improvements to crop yield, in particular for cereals, will be central to this challenging goal. 

Loss of wheat yield to disease is a major contributor to concerns surrounding global food security. In the UK and Europe, one disease dominates concern in the market, due to its destructive nature and widespread distribution [4]. The pathogen *Zymoseptoria tritici* (formerly *Septoria tritici*, also *Mycosphaerella graminicola*), which causes Septoria leaf blotch (SLB), thrives throughout the wheat growing regions of the world, and can cause loss of wheat yields above 30% [5]. 

The unusual infection cycle of *Z. tritici* may be a contributory factor to the success of this pathogen. Penetration of the plant is exclusively via the stomata [6]; once inside the plant cells, there is a long symptomless phase. This phase is seen in other species of the Mycosphaerella genus and is not common amongst other plant pathogens [7]. This long biotrophic stage of SLB typically lasts for 10–20 days [7,8], and does not cause the plant to present with any symptoms, and thus infected plants cannot be visually identified at this early infected stage. Once the pathogen becomes necrotrophic and symptoms appear, the infection is already well established, with the production of reproductive conidia following closely, around 20 days post infection [9]. Thus, an outbreak of SLB can be established in a field before it can be identified, and fungicide can be applied in an attempt to reduce its spread. As such, understanding how the pathogen evades host defences during its biotrophic stage will be an essential part of a combined effort in controlling the disease. 

It has been shown that plant interactions with pathogens, including *Z. tritici*, are altered under differential nitrogen availability [10], and specifically that plants grown under conditions of high nitrogen availability appear to be more susceptible to the disease [11,12,13]. Nitrogen availability is an essential factor in the cultivation of high yielding crops; therefore, it is not practical to grow wheat under reduced nitrogen availability to counter SLB. However, if we can understand the mechanisms which lead to increased tolerance under such conditions, we may be able to elucidate a novel approach to increasing SLB resistance in the field. Understanding this link starts with understanding the transcriptional reprogramming which occurs when the plant is grown under these stresses, and determining what links them.

WRKYs are a large family of well conserved, plant-specific transcription factors, responsible for altering the expression of other genes [14]. WRKY expression has been shown to be differentially regulated in response to drought, cold, and salinity [15,16,17], as well as by infection with *Agrobacterium*, powdery mildew, bacterial wilt, and rice blast [18,19,20]. Family-wide changes in expression have been seen for WRKYs in response to Septoria [21,22,23], although an analysis of changes in expression of individual WRKYs is lacking. In response to abiotic stress including drought, cold and salinity, 41 of the 103 WRKYs in rice were shown to be differentially expressed [24]. The knockout of *WRKY3* and *WRKY4* expression in Arabidopsis has been shown to result in susceptibility to pathogens such as *Botrytis cinera* [25], whilst the overexpression of a *GmWRKY12* and *TaWRKY2* has been shown to increase resistance to drought in soybean and wheat, respectively [26,27]. It is apparent that this family of transcription factors, one of the largest in the plant kingdom [28], is central to the response of plants to both abiotic and biotic stress. Furthermore, preliminary work for this project identified the WRKY transcription factor family as being of particular interest, noting their differential expression in response to varying nitrogen availability (Appendix A). WRKYs also demonstrate an ability to “cross-talk”—that is, to regulate the expression of other WRKYs. This is indicated by the enrichment of WRKY promotor regions with “W-boxes”. W-boxes are a short motif (TGGAC[C/T]), known to be the binding site of WRKY transcription factors. Therefore, those genes with enriched W boxes in their promoter regions are likely to be under the transcriptional regulation of WRKYs. The universal involvement of WRKYs in a wide range of stress responses, combined with their ability to cross-regulate expression of other WRKYs, makes them ideal candidates to control the interaction between nitrogen and SLB.

A quantitative real-time polymerase chain reaction (qPCR) was used to assess gene expression of WRKY transcription factors in wheat under different stress conditions. This technique is a fast, cheap method for analysing the expression of specific genes, which requires only part of the gene’s coding sequence to be known. This allowed us to determine the relative expression of members of the WRKY transcription factor family, which had previously been identified as showing differential expression in response to nitrogen stress and were therefore of interest.

Septoria abundance is typically measured using visual disease score. However, by its very nature, this is subjective, and therefore may be unreliable and difficult to replicate. This is particularly true when trying to assess Septoria, where infected plants may be asymptomatic for several weeks. Unless the disease has progressed to its necrotrophic phase, visually scoring the disease is not possible. Therefore, quantitative PCR was also used to assess the total amount of fungus biomass in the plant, therefore indicating disease severity in an infected sample [29]. The *ITS* region of Septoria is highly specific—it was used in the reclassification of *Z. tritici* into a separate genus from other Septoria species [30,31]. Therefore, it is an ideal candidate for quantitatively assessing the amount of Septoria present within an infected sample. By comparing the abundance of *ITS* in infected plants to one another, we present a novel relative quantitative measure of Septoria abundance in wheat.

## 2. Results

### 2.1. Septoria Abundance

Elongation factor 1α (EF1α) was validated as an endogenous control, as it showed non-significant differences CT values between the six treatment conditions (ANOVA, *p* = 0.478, data not shown). Quantitative real time PCR showed that the *Zymoseptoria tritici* ITS gene increased in abundance with increasing Septoria symptoms (Figure 1A,B. Wheat assessed as having very low levels of Septoria symptoms was used as the reference condition, with expression of ITS increasing 2.5 fold for medium Septoria levels and over five-fold higher in high levels of Septoria. Differences in abundance were highly significantly different in each treatment level (ANOVA with Tukey post-hoc, *p* < 0.001, *n* = 3).

The growth of wheat was shown to be inhibited under severely reduced nitrogen availability, as determined by plant surface area estimate. Plants grown under 5.25 mM nitrate were not smaller after 5 weeks than those grown under 7.5 mM nitrate, but plants grown under 2.25 mM nitrate were 32% (*p* < 0.001) smaller than controls.

Despite differences in surface area, however, plants grown under severely reduced nitrogen availability did not show significantly differential fungal load, as determined by qPCR (Figure 1C, *p* > 0.05). Disease severity could not be assessed visually or by spore wash, as the disease was still in its biotrophic phase, and thus visual symptoms were not present.

Fungal load did not correlate with plant surface area after 5 weeks growth (Figure 1D, *p* = 0.628).

### 2.2. WRKY Expression

An initial microarray screen identified the WRKY transcription family as having highly altered expression in response to reduced nitrogen availability (Appendix A). Indeed, WRKY family members demonstrated differential upregulation and downregulation of expression under these conditions, and as such they were identified as candidates to link nitrogen stress response to other stressors. In contrast, there was no detectable differential expression in the bZIP, MYB or bHLH transcription factor families. To investigate which specific WRKYs were responding to nitrogen stress, and elucidate whether these were also differentially expressed in response to *Septoria*, qPCR primers were designed against the WRKYs, for which a known coding sequence was publicly available at the time. The expression of those WRKYs with detectable levels of mRNA was assessed under nitrogen and *Septoria* stress individually (Appendix A), and a candidate list derived based on WRKYs which responded strongly to either nitrogen, *Septoria* or both, and for which reliable transcript abundance information could be obtained, as designated by a single-peaked melt curve and single PCR product.

Six WRKY genes (*WRKYs 2*, *10*, *19*, *39*, *53b* and *68a*) were investigated for their differential expression in response to reduced nitrogen input. All six genes were seen to be differentially expressed (ANOVA with Tukey post-hoc, *n* = 3, *p* < 0.05) in plants grown under reduced nitrogen for five weeks prior to infection (Figure 2).

*WRKY2*, *WRKY53b* and *WRKY68a* showed the upregulation of expression under both moderately and severely reduced nitrogen, whilst *WRKY19* showed upregulation only under severe levels of nitrogen stress. *WRKY10* and *WRKY39* showed significant upregulation under moderately reduced nitrogen, but changes under severe nitrogen were not deemed to be statistically significant. The only gene to show a significant (*p* < 0.05), dose-dependent response was *WRKY53b*—a 1.4 fold upregulation under moderate nitrogen stress, increasing to a 2.5 fold upregulation under 2.25 mM nitrogen. The expression of *WRKY2* and *WRKY68a* did not significantly change between 5.25 mM and 2.25 mM nitrogen conditions.

Expression of these WRKY genes was assessed under Septoria infection. Samples were scored for Septoria severity as either very low, low or medium according to the amount of leaf surface covered by symptoms of SLB. Of the six genes analysed, three were differentially expressed in response to both low and medium levels of Septoria (Figure 3). The differential expression of *WRKY39* was of greatest interest, which showed a significant (*p* < 0.001), dose-dependent downregulation, with very low levels of Septoria showing a 15 fold downregulation, and low levels showing greater than 35-fold downregulation. It was the only WRKY to be significantly downregulated under Septoria infection. *WRKY68a* also showed a dose-dependent response with significant (*p* < 0.05) levels of upregulation, with three-fold upregulation seen under very low levels of Septoria, and greater than five-fold upregulation seen under low levels. The expression of *WRKY2* and *WRKY10* were both significantly upregulated (*p* < 0.05), however the magnitude of their change in expression was smaller, both showing less than three-fold changes in both conditions.

The expression of three WRKY genes (*WRKY39*, *WRKY53b* and *WRKY68a*) was assessed in a combined stress assay to determine if an interaction existed between nitrogen and Septoria infection in determining the expression of specific WRKY transcription factors. Gene expression was analysed 30 days post infection, with the infection still in the biotrophic phase. Differential expression of all three WRKYs was seen as a response to both reduced nitrogen and Septoria infection, when compared to uninfected plants grown under 7.5 mM nitrogen (Figure 4A). This was expected, given the differential expression seen under single stress assays (Figure 2 and Figure 3). 

The qualitative changes in *WRKY39* and *WRKY53b* expression, when grown under reduced nitrogen, were the same, when compared to 7.5 mM nitrogen, whether they were infected with *Z. tritici* or not (Figure 4B). *WRKY39* was upregulated under 5.25 mM nitrogen and downregulated under 2.25 mM nitrogen, for both infected and uninfected plants, although the sale and significance of these was infection dependent. *WRKY53b* was upregulated under both 5.25 mM and 2.25 mM nitrogen, confirming results from the single stress assay. A two-way ANOVA analysis revealed that the interaction between nitrogen concentration and Septoria infection was non-significant for *WRKY39* (*p* = 0.101) and *WRKY53b* (*p* = 0.454), demonstrating that this qualitative observation is supported quantitatively. For *WRKY68a*, reduced nitrogen resulted in upregulation for non-infected plants. However, in plants infected with Z. tritici, *WRKY68a* did not show significant changes in expression when grown under reduced nitrogen. A two-way ANOVA shows that the interaction between nitrogen concertation and Septoria infection is significant (*p* = 0.010). This shows that the way in which *WRKY68a* expression changes in response to reduced nitrogen is dependent on whether the plant is infected with *Z. tritici* or not, or vice versa. 

This finding is perhaps more pertinently illustrated by analysing the change in gene expression in response to Septoria infection for plants grown under each nitrogen condition separately (Figure 4C). By comparing how WRKY expression changes in response to Septoria when grown under 7.5 mM, 5.25 mM and 2.25 mM nitrogen, we can see that changes in *WRKY53b* expression are similar across all nitrogen concentrations. This is supported by a one-way ANOVA comparing the ΔΔCT values, the differences in which are not significant (*p* = 0.275). Changes in expression of WRKY68a, however, show a significant treatment effect (*p* = 0.002) between nitrogen treatments, confirming that changes in *WRKY68a* expression in response to Septoria are dependent on the nitrogen concentration. Changes for *WRKY39* expression were close to significance (*p* = 0.064), and so may merit further investigation. WRKY39 was the only gene to show significant downregulation when infected with Septoria under all three nitrogen treatments.

The interactions of *WRKY68a* were investigated using the STRING database. The fifty interactions with the highest confidence value were selected for plotting (Figure 5), plus fifty “second level” interactions—that is, the interactions of proteins which interact with WRKY68a. This allowed a more complete, better-connected network to be established. All interactions had a confidence score of or above 0.542, where 0.4 represents “medium confidence” and 0.7 represents “high confidence”. WRKY68a is shown interacting with several major groups of proteins known to be involved in biotic and abiotic stress responses, including other WRKYs, calmodulin binding transcription activator (CAMTA) factors, MAP kinases, zinc homeodomain proteins and ethylene response factors.

## 3. Discussion

It has previously been hypothesized that plant surface area mediated the observed link between Septoria leaf blotch susceptibility and nitrogen input, due to the stochastic nature of the fungus’ spread. Therefore, it was necessary to assess the link between nitrogen and plant surface area and nitrogen, and plant surface area and Septoria, to determine if this factor may act as an intermediary in this link. Our results show that leaf surface area is not correlated with fungal load (Figure 1D), regardless of the size of the plant, thus suggesting that this factor alone cannot explain observed differences in Septoria severity.

Our results indicate that disease severity did not change when wheat was grown under reduced nitrogen conditions for 5 weeks prior to infection. Although mean disease severity did decrease slightly as nitrogen concentration dropped (Figure 1C), these changes were not statistically significant. Plants in the long-term nitrogen assay also did not show any significant differences in disease severity between nitrogen treatments. Therefore, it may be suggested that disease severity is not linked to nitrogen input during the biotrophic phase of the infection, regardless of the stage of plant growth.

This is in contrast to previous observations, where increased nitrogen was found to result in increased Septoria severity [11,13,32]. However, these previous observations have been based on visual assessments of disease severity. As discussed previously, SLB only presents with symptoms during the necrotrophic phase [7], and therefore previous assessments have only been possible during this latter phase of infection. By assessing fungal load using qPCR, it was possible to assess disease severity in the earlier, biotrophic phase of the infection. Therefore, the data suggest that observed increases in Septoria susceptibility seen under increased nitrogen availability are only present in the necrotrophic phase of the infection, representing an important finding in developing our understanding of Septoria leaf blotch progression. 

We have demonstrated that quantitative PCR could be used to provide a reliable, reproducible and unbiased method for assessing relative disease severity in wheat. Samples from uninfected tissue showed no amplification of Septoria transcripts, confirming their specificity to diseased tissue. However, this made uninfected tissue unsuitable as a reference condition for the ΔΔC_T_ method of relative gene expression analysis—some level of expression is required to provide a reference. Therefore, the very low disease severity was used as a reference condition and assigned a value of 1, and an abundance of *ITS* in other samples was compared to the expression of *ITS* in the very low Septoria samples. Primers for the elongation factor 1 α subunit (*EF1*α) gene were used as an endogenous control. Because *ITS* showed a quantitative increase as disease severity increased, it was concluded that the abundance of *ITS* represented a reliable, quantitative method of assessing disease severity, which does not rely on the disease producing visible symptoms. Therefore, we present quantitative PCR as a novel method for assessing the effect of nitrogen availability on Septoria tolerance, at an earlier stage than was previously possible. Disease severity is measured as an expression of *ITS* normalized against expression of *EF1α*. It should be noted that disease severity using this method is relative to a reference; in order to produce an absolute measure of fungal load, it will be necessary to analyse *ITS* abundance against known masses of *Zymoseptoria tritici*.

Differential expression of a range of *WRKYs* was seen under reduced nitrogen conditions (Figure 2). *WRKY19* was over 2-fold upregulated under 2.25 mM nitrogen compared to 7.5 mM. However, under moderately reduced nitrogen (5.25 mM), no differential expression was seen, suggesting that it plays a role in the plant response to more severe stress levels. Expression of *WRKY68a* was over two-fold higher under moderate nitrogen stress, but its expression did not further increase under severe nitrogen stress. This suggests that the role played by WRKY68a, if linked to a nitrogen stress response, corresponds to less dramatic changes in wheat nitrogen availability. It is possible, therefore, that *WRKY68a* may be expressed as part of a generic stress response, as the severity of the stress does not appear to affect its expression. Whatever their precise roles, both *WRKY19* and *WRY68a* appear to show a threshold of nitrogen stress at which their expression is upregulated, suggesting that the response may be an on/off signal. In contrast to both of these, *WRKY53b* expression exhibited a dose-dependent increase in response to nitrogen decrease. Under moderate stress, expression of *WRKY53b* increased 1.43 fold, whilst under severe nitrogen stress, this increased to over 2.5 fold. Therefore, it is possible that *WRKY53b* mediates the response of wheat to the quantitative effects of reduced nitrogen, possibly controlling key growth parameters. 

Data from the combined stress assay showed that *WRKY39* was also downregulated in the biotrophic stage of the infection cycle, across all three nitrogen concentrations. It is apparent that infection with *Z. tritici* results in the downregulation of expression of *WRKY39* throughout the infection cycle, suggesting that *WRKY39* may play an important role in the defence of wheat against this pathogen. It is unlikely that the suppression of *WRKY39* expression is via the traditional route of pathogen effectors. If a pathogen is able to alter host gene expression via effectors which have been secreted intracellularly, it would enable the plant to recognize the pathogen via the guard hypothesis [33]. As pathogen recognition is not thought to occur during the biotrophic stage, it is unlikely that this is the mode of WRKY suppression. As little is known about the molecular mechanisms employed by Septoria to evade host detection during this phase, however, it is difficult to hypothesize using the mechanism by which this occurs. However, host processes such as senescence have been shown to be altered even during the asymptomatic phase [34], and thus WRKY39 may play a role in such processes. Nevertheless, it is likely that this WRKY controls an important interaction in determining the success of the fungus, and as such, requires further investigation.

*WRKY68a* showed the largest increase in expression under necrotrophic phase SLB, showing over 5-fold upregulation in the plants with a medium Septoria score compared to those deemed very low (Figure 3). However, *WRKY68a* does not show the same 5-fold upregulation during the biotrophic phase as it did during the necrotrophic phase (Figure 4C). Instead, under 7.5 mM nitrogen, no differential expression is seen. This suggests that WRKY68a may be important in the defence response against Septoria in the necrotrophic phase, once the pathogen has been identified as such by the plant. 

In assessing the effect of combined nitrogen and Septoria stress, *WRKY39* was significantly upregulated in infected plants when grown under 5.25 mM nitrogen compared to 7.5 mM, but no other reduced nitrogen conditions produced significant changes in gene expression. Importantly, a two-way ANOVA showed a non-significant interaction term between nitrogen and infection status (*p* = 0.101). This suggests that the way in which *WRKY39* expression changes in response to reduced nitrogen is the same for infected plants and uninfected plants. Similarly, only moderately reduced nitrogen (5.25 mM) resulted in the significant downregulation of *WRKY53b* expression in infected tissue, and no changes in uninfected plants were significant in response to reduced nitrogen. Again, the interaction term in a two-way ANOVA between nitrogen and infection status was non-significant (*p* = 0.454). For both of these genes, this non-significant interaction term confirms a qualitative assessment—the patterns of gene expression appear to be the same when nitrogen is reduced for both infected and non-infected plants.

*WRKY68a* also shows only one condition in which nitrogen reduction causes significant differential expression in either infected or uninfected plants. Uninfected plants grown under 5.25 mM nitrogen show significantly higher *WRKY68a* expression than those grown under 7.5 mM nitrogen. However, the interaction term between nitrogen and infection status is significant (*p* = 0.010). This suggests that the way in which WRKY68a expression changes in response to reduced nitrogen is dependent on whether or not the plant is infected with Septoria. Again, this is consistent with a qualitative assessment. For uninfected plants, a 2.65-fold upregulation of *WRKY68a* expression is seen under 5.25 mM nitrogen, and a smaller, albeit statistically non-significant, upregulation of 1.28-fold is seen under 2.25 mM nitrogen. However, for plants infected with Septoria, the mean change in gene expression under reduced nitrogen is a downregulation of expression, although, again, neither show statistical significance compared to 7.5 mM nitrogen. Given the expression pattern of *WRKY68a* shown in the study, and the fact that it has been shown to confer multiple stress tolerance when expressed in Arabidopsis [18], it is an ideal candidate for linking the two stressors. In order to investigate this further, we interrogated the proteins which interact with *WRKY68a* using the STRING database (Figure 5). By visualizing the network of interacting proteins, we are able to hypothesize a possible molecular basis for the predicted link between nitrogen stress and Septoria tolerance. The WRKY68a protein is shown to interact with calmodulin transcription activation (CAMTA) factors, a family of proteins known to be involved in abiotic stress response [35]. *WRKY68a* is also shown to interact with ethylene response factors (ERFs), a family known to be involved in the response to a wide variety of pathogens [36]. Therefore, we propose that *WRKY68a* may act as a link between these families of proteins, and coordinate the stress responses. Additionally, *WRKY68a* interacts with MAP kinase proteins. The MAPK pathway is central to stress responses, both biotic and abiotic [37], and therefore it may be via an interaction with the MAP kinase pathway that *WRKY68a* coordinates the nitrogen and Septoria stress responses in wheat. In order to investigate these proposed links and the roles played by other WRKYs further, differential expression assays should be performed. These can either be in the form of transgenic overexpression in wheat [26], or using the high functional conservation of WRKYs to express these transcription factors exogenously in other species [38]. Information gained from these experiments can be used to provide a screening program for the optimal expression of molecular breeding markers. Such markers may be mapped to current or novel quantitative trait loci (QTLs), or may inform breeding strategies to produce an optimum combination of WRKY expression which provides the best resistance to Septoria.

It should be noted that WRKY transcription factors other than *WRKY68a*, *WRKY53b* and *WRKY39* were differentially expressed in response to nitrogen and Septoria infection (Appendix A), but were not included for subsequent analysis. Given the complex and interconnected nature of WRKY transcription factor interactions, a role for other WRKYs in the coordination of the nitrogen and Septoria stress responses should not be ruled out. 

That *Zymoseptoria tritici* abundance can be quantified in wheat without the need for visual symptoms is an important finding, and the qPCR method presented above may be useful in understanding the early stages of this devastating disease. We present a range of *WRKY* transcription factors as being differentially expressed in response to reduced nitrogen and Septoria in isolation and in combination, and hypothesize that the co-dependent nature of *WRKY68a* expression under these stressors suggest that this transcription factor may play a role in coordinating these stress responses via its interaction with CAMTA transcription factors, ERFs and the MAP kinase pathway.

## 4. Materials and Methods 

### 4.1. Plant Growth Conditions and Physiological Assessment

Winter wheat (*Triticum aestivum* var: Cordiale) was grown from seed in silver sand with one seedling per pot (*n* = 8 for each nitrogen treatment). Plants were provided with nutrients via modified Hoagland’s solution. Nitrogen control plants (grown under 7.5 mM nitrogen) were supplied with 2.5 mM Ca(NO_3_)_2_·4H_2_O, 2.5 mM KNO_3_, 1 mM KH_2_PO_4_, 1 mM MgSO_4_·7H_2_O, 23 µM H_3_BO_3_, 4.6 µM MnCl_2_·4H_2_O, 0.38 µM ZnSO_4_·7H_2_O, 0.10 µM CuSO_4_·5H_2_O, 0.27 µM NaMoO_4_·2H_2_O, 14 µM FeEDTA. Moderately nitrogen stressed plants (70% of control, 5.25 mM nitrogen) were supplied with 1.75 mM Ca(NO_3_)_2_·4H_2_O, 0.75 mM CaCl_2_, 1.75 mM KNO_3_, 0.375 mM K_2_SO_4_, with all other nutrients the same as the control. Severely nitrogen stressed plants (30% of optimum, 2.25 mM) were supplied with 0.75 mM Ca(NO_3_)_2_·4H_2_O, 1.75 mM CaCl_2_, 0.75 mM KNO_3_, 0.875 mM K_2_SO_4_, with all other nutrients the same as the control. Temperature was maintained at 25^f^ with a 16h: 8h light:dark cycle. Humidity was maintained above 60% to maximize infection rates.

The effect of nitrogen was assessed after 35 days growth (*n* = 8), at which point plants were between growth stages (GS) 20 and 23 [39]. Leaf length and leaf width at its widest point were measured and an estimate of leaf surface area was obtained by taking the product of these. The total leaf surface area of the plant was calculated by finding the product of the estimated leaf size and the number of leaves on the plant.

### 4.2. Septoria Infection

Five plants were infected after physiological measurements were taken, with three plants left uninfected to act as controls. Infection was induced by spraying leaves with 2 mL *Zymoseptoria tritici* spore solution at a concentration of 5 × 10^5^ spores mL^−1^ in distilled water with 0.1% Tween20. Nutrients, temperature, light and humidity were maintained as they were previously. Additionally, plants were sprayed with dH_2_O until run off prior to infection, and then sprayed lightly every 48 h, as leaf wetness had been shown to positively influence Septoria infection [9]. Plants were observed for symptoms of Septoria. After 30 days, each plant was scored 0-9 for Septoria [40], and spore washes were taken from three representative leaves by removing the leaves, immersing in distilled water with 0.1% Tween20 and shaking vigorously for 1 min. All remaining above-ground tissue was harvested and snap frozen.

### 4.3. Tissue Collection from Field Samples

Leaf tissue was taken from field sites at Nafferton Farm (Nafferton Ecological Farming Group, Stocksfield, Northumberland; www.nefg-organic.org). Plants were scored for Septoria based on % disease coverage of leaf surface (<2%, 3–10%, 10–24% coverage scored as very low, low and medium respectively) and frozen in liquid nitrogen.

### 4.4. RNA Extraction and cDNA Synthesis

RNA was extracted from homogenized frozen tissue using PureLink RNA Mini Kit (ThermoFischer, Waltham, MA, USA). The kit is suitable for extracting both plant and fungal RNA from tissue samples. Reverse transcription was performed using Bioline SensiFast™ cDNA synthesis kit (Meridian Biosciences, Cincinatti, OH, USA), to produce cDNA for gene expression analysis.

### 4.5. Primer Design

Primers (Appendix A) were designed to amplify a section of the *Zymoseptoria tritici* Internal Transcribed Spacer (ITS) region of ribosomal DNA; a section of the *Z. tritici* Elongation Factor 1 subunit α (EF1α) mRNA, and sections of WRKY transcription factors. All primers were aligned against both wheat and *Z. tritici* genomes to ensure specificity (BLAST, https://www.nih.gov). 

### 4.6. qPCR

Quantitative real-time PCR was performed using a QIAGEN RotorGene 6000 (QIAGEN, Venlo, The Netherlands) in 15 µL reactions, using Bioline 2X SensiFAST™ SYBR^®^ No-ROX Kit Master Mix (Meridian Biosciences, Cincinatti, OH, USA), with primers at 200 µM final concentration. Three technical replicates were performed per sample. The PCR conditions were 60 s of denaturing at 95 °C, followed by 40 cycles of 5 s denaturing at 95 °C and 10 s annealing/extension at 60 °C, with fluorescence acquisition at the end of every annealing/extension step. A melt curve was performed at the end of the run, to evaluate the amplicon by increasing the temperature by 1 °C from 60 °C to 95 °C.

### 4.7. Data Analysis 

Relative Septoria load and relative WRKY expression were assessed using a modified ΔΔC_T_ method [41]. ΔC_T_ values were calculated using the *Z. tritici* elongation factor 1 subunit α gene and wheat ubiquitin gene, respectively, as endogenous controls (Appendix A). Both control genes were validated as appropriate. A reference condition pertinent to the respective experiment was chosen to calculate ΔΔC_T_ values and subsequently elucidate a relative fold change. 

Septoria abundance and WRKY single stress assays were analysed for significantly different ΔC_T_ values using one-way ANOVA. If a significant treatment effect (*p* < 0.05) was observed, then a Tukey’s post-hoc was used to determine which conditions showed significantly different values. Dual stress assays were analysed using a two-way ANOVA, to determine the statistical significance of differences between treatments and the interaction between them. 

## Figures and Tables

**Figure 1 ijms-21-04165-f001:**
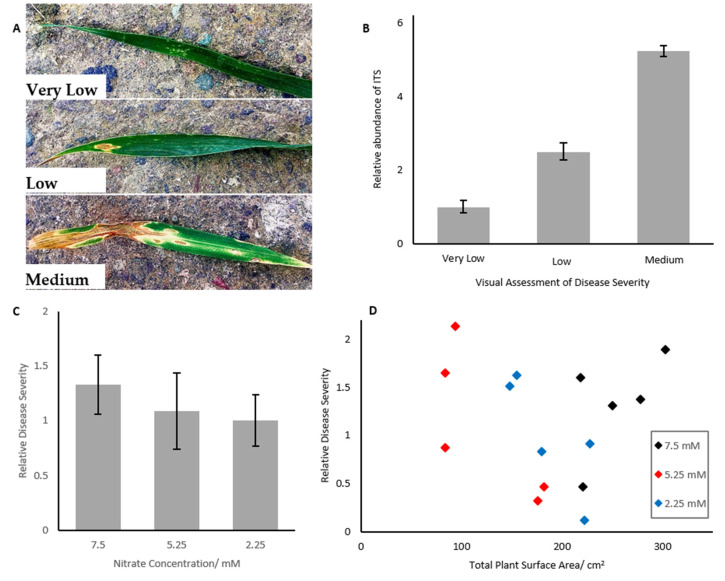
(**A**) Representative images of Very Low, Low and Medium levels of Septoria infected wheat leaves. (**B**) Abundance of Septoria *ITS* region in wheat samples infected with Septoria. *ITS* abundance increases with increasing disease severity. Error bars represent ± 1 standard deviation. *n* = 3. Different letters represent statistically significantly different means (*p* < 0.05). Means compared via ANOVA, *n* = 3 (**C**) Wheat grown for 35 days under differential nitrogen concentrations did not show significantly different levels of Septoria (*p* < 0.05), analysed by expression of the Septoria ITS region. Error bars represent ± 1 SEM. Different letters represent statistically significantly different means (*p* < 0.05). Means compared via ANOVA, *n* = 5. (**D**) Septoria severity in wheat did not correlate with total surface area of the plant for wheat grown for five weeks under differential nitrogen treatment. Regression analysis was used to determine correlation, and ANOVA was used to determine that the slope coefficient was not significantly different from 0 (*p* = 0.682).

**Figure 2 ijms-21-04165-f002:**
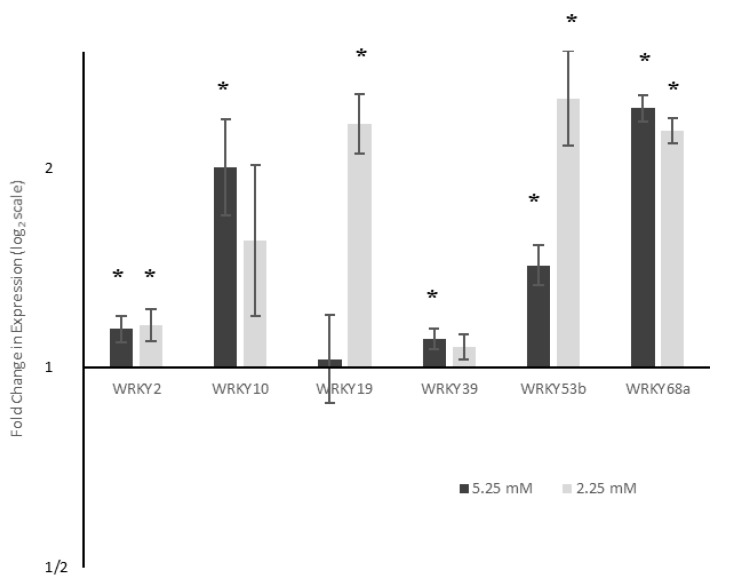
Expression of six WRKY transcription factors under reduced nitrogen availability. Gene expression is shown relative to expression levels when grown under 7.5 mM NO_3_^−^. ΔΔC_T_ values were analysed using ANOVA followed by Tukey post-hoc, *n* = 3. * *p* < 0.05. Expression of all WRKYs was significantly (*p* < 0.05) upregulated under at least one reduced nitrogen condition.

**Figure 3 ijms-21-04165-f003:**
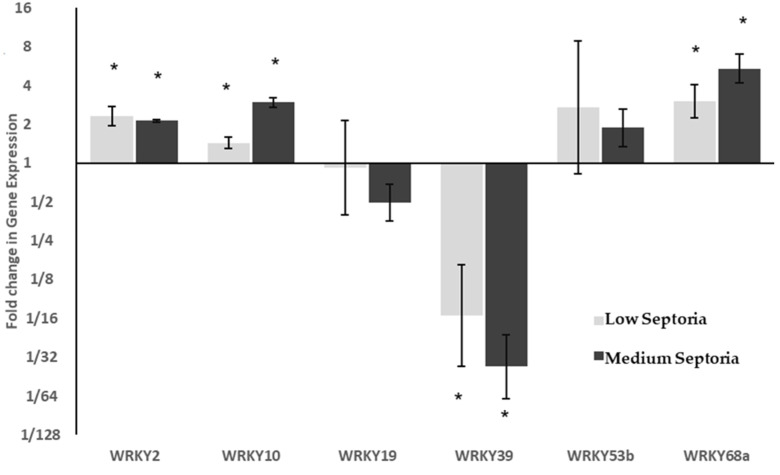
Relative expression of WRKY transcription factors in response to Septoria Leaf Blotch. ΔC_T_ values were analysed using ANOVA followed by Tukey post-hoc. * *p* < 0.05 in post-hoc analysis. Changes in expression were compared to plants showing Very Low symptoms of SLB. WRKY39 shows significant downregulation, whilst *WRKY68a* shows a dose dependent increase in expression with increasing SLB severity.

**Figure 4 ijms-21-04165-f004:**
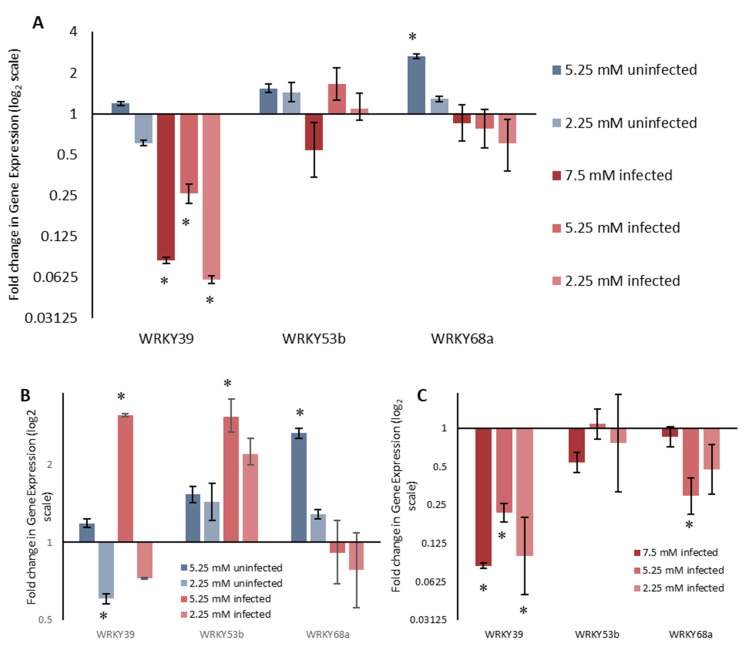
Expression of three *WRKY* transcription factors under combined stress assay of reduced nitrogen availability and biotrophic stage Septoria infection. (**A**) Fold change is displayed relative to plants grown under 7.5 mM nitrogen, which were not infected with Septoria. (**B**) Fold change is displayed relative to plants grown under 7.5 mM in the same infection state. (**C**) Differential expression of three *WRKYs* caused by infection with Septoria when grown under different nitrogen treatments. Fold change is displayed relative to uninfected plants grown at the same nitrogen concentration. ALL ΔC_T_ values were analysed using ANOVA followed by Tukey post-hoc, *n* = 6. * *p* < 0.05 for difference in ΔC_T_ value compared to reference condition.

**Figure 5 ijms-21-04165-f005:**
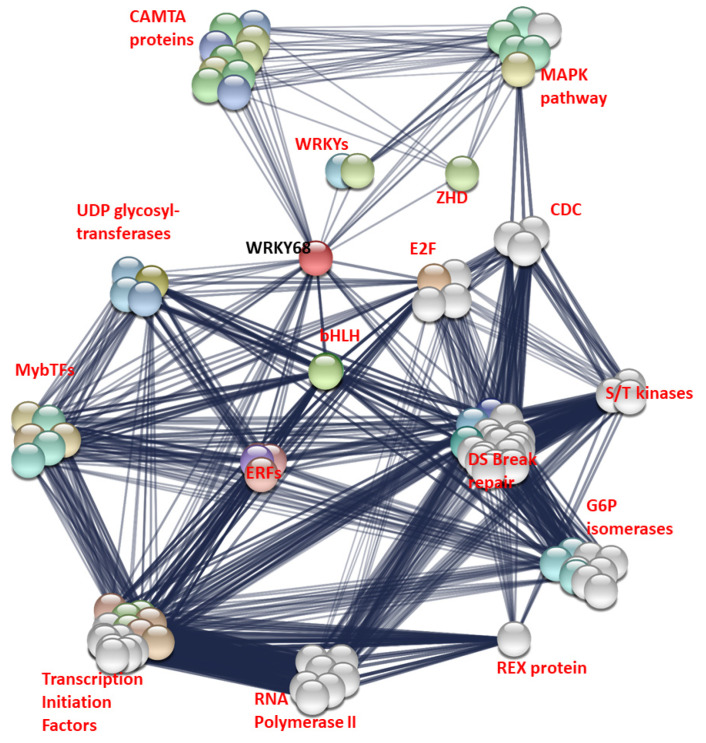
Interaction map of WRKY68a. Coloured nodes represent proteins known or predicted to interact with WRKY68a. White nodes represent proteins known to interact with those proteins. Edges represent a known or predicted interaction. The thickness of the edge indicates the strength of evidence for the interaction. Proteins are grouped into families or pathways for simplicity. CAMTA = calmodulin transcription activation factors, MAPK = nitrogen activated protein kinase, ZHD= zinc finger homeobox domain proteins, CDC = cell division cycle proteins, S/T = serine/threonine, bHLH = basic helix-loop-helix, ERFs = ethylene response factor, DS Break repair = double strand break repair proteins, including RAS51, FHA, NBS1 and BRCA proteins.

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
