# Peer review of "Septoria Leaf Blotch and Reduced Nitrogen Availability Alter WRKY Transcription Factor Expression in a Codependent Manner"

_ijms, 2020, doi:10.3390/ijms21114165_

Round 1
Reviewer 1 Report
The addition of Figure 5 was a significant improvement to the manuscript
Reviewer 2 Report
The resubmitted manuscript by Poll et al. has been conveniently revised, and major previous concerns addressed.
Minor points:
- As additional WRKY factors (for example, WRKY8) showed de-regulation upon nitrogen/fungi treatments, you should include in the abstract/conclussions that apart WRKY68a other WRKY factors could also be mediating the interaction nitrogen/defence.
- Supplemental Figure 4 should be revised (error bars, legend).
Author Response
Please see the attachment

This manuscript is a resubmission of an earlier submission. The following is a list of the peer review reports and author responses from that submission.
Round 1
Reviewer 1 Report
Nice work regarding the potential to detect the causative agent of Septoria leaf blotch in wheat prior to visual detection. Perhaps can be used as a surveillance method under suspect conditions.
Some minor edits:
Line 12 in abstract and line 339: Italicize " Zymoseptoria tritici"
Line 102: ".. PCR showed that the ITS gene increased.." since the M&M are at the conclusion of the manuscript identify that it is Z. tritici ITS gene
Line 138 reads "The expression of WRKY2 and WRKY51 were both significantly upregulated.." Do the authors mean WRKY2 and WRKY10? There is no WRKY51 elsewhere in the manuscript.
Fig: 1D can the legend box be placed elsewhere or boxed - it can be mistaken for data points as it is currently.
Author Response
Thank you for your comments on this manuscript. We have addressed your comments in our revised manuscript, the details of which are addressed below.
Point 1:
Line 12 in abstract and line 339: Italicize " Zymoseptoria tritici"
Line 102: ".. PCR showed that the ITS gene increased.." since the M&M are at the conclusion of the manuscript identify that it is Z. tritici ITS gene
Line 138 reads "The expression of WRKY2 and WRKY51 were both significantly upregulated.." Do the authors mean WRKY2 and WRKY10? There is no WRKY51 elsewhere in the manuscript.
Fig: 1D can the legend box be placed elsewhere or boxed - it can be mistaken for data points as it is currently.
Response 1: All suggested corrections made as above.

Reviewer 2 Report
The paper documents a correlation between changes in expression of members of the WRKY TF factors and Septoria/N and is well constructed and the fact that not all the WRKYs respond or change in the same direction provides a nice internal control for the study. Figure 4 is particularly nice
The study is not ground-breaking but provides a valid addition to the wheat literature. The authors do not provide any gene network model to link Septoria response to N and how WRKY68a could moderate one or both of these factors and this would have been appreciated. Its not clear how the study provides a pathway to new markers for breeding and the real potential is in the level of understanding the biology, particularly with a good wheat genome assembly now being available.
Author Response
Thank you for your comments on this manuscript. We have addressed your comments in our revised manuscript, the details of which are addressed point by point below.
Point 1: The authors do not provide any gene network model to link Septoria response to N and how WRKY68a could moderate one or both of these factors and this would have been appreciated.
Response 1: In order to address this point we have included a proposed protein interaction network for WRKY68a as an additional figure (Fig 5). Accordingly we have modified the text in our results section as follow, lines 176-184:
“The interactions of WRKY68 were investigated using the STRING database. The fifty interactions with the highest confidence value were selected for plotting (Fig 5), plus fifty “second level” interactions – that is the interactions of proteins which interact with WRKY68a. This allowed a more complete, better connected network to be established. All interactions had a confidence score of or above 0.542, where 0.4 represents “medium confidence” and 0.7 represents “high confidence”. WRKY68 is shown interacting with several major groups of proteins known to be involved in biotic and abiotic stress responses (REF), including other WRKYs, calmodulin binding transcription activator (CAMTA) factors, MAP kinases, zinc homeodomain proteins and Ethylene Response Factors.”
We have also added the following text discussing the implications of these results in the discussion section, lines 288-299:
“In order to investigate this further, we interrogated the proteins which interact with WRKY68a using the STRING database (Fig. 5). By visualising the network of interacting proteins like this we are able to hypothesise a possible molecular basis for the predicted link between nitrogen stress and Septoria tolerance. WRKY68a is shown to interact with calmodulin transcription activation (CAMTA) factors, a family of proteins known to be involved in abiotic stress response [35]. WRKY68a is also shown to interact with ethylene response factors (ERFs), a family known to be involved in the response to a wide variety of pathogens [36]. Therefore we propose that WRKY68a may act as a link between these families of proteins, and coordinate the stress responses. Additionally, WRKY68a interacts with MAP kinase proteins. The MAPK pathway is central to stress responses both biotic and abiotic [37], and therefore it may be via an interaction with the MAP kinase pathway that WRKY68a coordinates the nitrogen and Septoria stress responses in wheat.”
We have modified the text of the materials and methods section by including the following, lines 377-384:
“4.8 Protein interaction analysis
The accession code for TaWRKY68a was searched against the STRING database (https://string-db.org). An interaction network was generated with the following settings: minimum confidence - medium (0.400), meaning of network edges – confidence (line thickness indicates the strength of data support), 1st shell – no more than 50 interactions, 2nd shell no more than 50 inteactions, display simplifications – “disable structure previews inside network bubble” and “hide node labels”. The nodes in the map were rearranged into protein families and pathways in order to simplify the network.”
This has allowed us to suggest a molecular basis for how N and Septoria responses may be linked via WRKY68a, by linking families of proteins known to play a role in stress responses to abiotic (CAMTA factors, MAP kinase pathway) and biotic stress (ERFs, MAP kinase pathway).
Point 2: Its not clear how the study provides a pathway to new markers for breeding and the real potential is in the level of understanding the biology, particularly with a good wheat genome assembly now being available
Response 2: We have clarified the routes by which information gained from these experiments can be translated into improved breeding strategies in our discussion (lines 303-306):
“Information gained from these experiments can be used to provide a screening programme for optimal expression of molecular breeding markers. Such markers may be mapped to current or novel quantitative trait loci (QTLs), or may inform breeding strategies to produce an optimum combination of WRKY expression which provide the best resistance to Septoria”

Reviewer 3 Report
The manuscript by Poll et al. tries to determine the relationships between Septoria Leaf Blotch, reduced nitrogen availability and WRKY expression. Although the manuscript presents valuable results, they seem very weak to support the main conclusion of the manuscript. The WRKY transcription factors are a very large family with many members acting in the same pathways. Thus, the detected variations in the mRNA levels of WRKY68a are not a sufficient proof to hypothesise a role of this TF in coordinating Septoria and reduced nitrogen responses. As authors mention in the discussion section, additional experiments are necessary to verify the role of a specific gene in the plant (i.e. those obtained from silencing or overexpressing wheat lines).
Author Response
Thank you for your comments on this manuscript. We have addressed your comments in our revised manuscript, the details of which are addressed below. We hope to have clarified the difference between our conclusions from our data, and our subsequent hypotheses which require further testing.
Point 1: The manuscript by Poll et al. tries to determine the relationships between Septoria Leaf Blotch, reduced nitrogen availability and WRKY expression. Although the manuscript presents valuable results, they seem very weak to support the main conclusion of the manuscript. The WRKY transcription factors are a very large family with many members acting in the same pathways. Thus, the detected variations in the mRNA levels of WRKY68a are not a sufficient proof to hypothesise a role of this TF in coordinating Septoria and reduced nitrogen responses. As authors mention in the discussion section, additional experiments are necessary to verify the role of a specific gene in the plant (i.e. those obtained from silencing or overexpressing wheat lines).
Response 1: We have included an additional figure (Fig 5) of WRKY68a protein interactions. Accordingly we have included this in our results section, lines 176-184
“The interactions of WRKY68 were investigated using the STRING database. The fifty interactions with the highest confidence value were selected for plotting (Fig 5), plus fifty “second level” interactions – that is the interactions of proteins which interact with WRKY68a. This allowed a more complete, better connected network to be established. All interactions had a confidence score of or above 0.542, where 0.4 represents “medium confidence” and 0.7 represents “high confidence”. WRKY68 is shown interacting with several major groups of proteins known to be involved in biotic and abiotic stress responses (REF), including other WRKYs, calmodulin binding transcription activator (CAMTA) factors, MAP kinases, zinc homeodomain proteins and Ethylene Response Factors.”
We have also added a paragraph discussing the implications of these results in the discussion section, lines 288-299
“In order to investigate this further, we interrogated the proteins which interact with WRKY68a using the STRING database (Fig. 5). By visualising the network of interacting proteins like this we are able to hypothesise a possible molecular basis for the predicted link between nitrogen stress and Septoria tolerance. WRKY68a is shown to interact with calmodulin transcription activation (CAMTA) factors, a family of proteins known to be involved in abiotic stress response [35]. WRKY68a is also shown to interact with ethylene response factors (ERFs), a family known to be involved in the response to a wide variety of pathogens [36]. Therefore we propose that WRKY68a may act as a link between these families of proteins, and coordinate the stress responses. Additionally, WRKY68a interacts with MAP kinase proteins. The MAPK pathway is central to stress responses both biotic and abiotic [37], and therefore it may be via an interaction with the MAP kinase pathway that WRKY68a coordinates the nitrogen and Septoria stress responses in wheat.”
We have added details of how this figure was dderived in the materials and methods section, lines 377-384
4.8 Protein interaction analysis
The accession code for TaWRKY68a was searched against the STRING database (https://string-db.org). An interaction network was generated with the following settings: minimum confidence - medium (0.400), meaning of network edges – confidence (line thickness indicates the strength of data support), 1st shell – no more than 50 interactions, 2nd shell no more than 50 inteactions, display simplifications – “disable structure previews inside network bubble” and “hide node labels”. The nodes in the map were rearranged into protein families and pathways in order to simplify the network.
This gives support of our hypothesis that WRKY68a may link Spetoria and nitrogen response, allowing us to suggest a molecular basis for how N and Septoria responses may be linked via WRKY68a. We have ensured that our choice of language makes the distinction between conclusions drawn from our results and hypotheses derived from these conclusions. Silencing or overexpression lines would provide more solid evidence for these hypotheses, but are beyond the scope of this paper.

Round 2
Reviewer 3 Report
I appreciate your valuable effort to distinguish between conclusions drawn from your results and hypotheses derived from these conclusions, as well as the attempts to support your hypothesis using STRING database.
Unfortunately, this approximation is very weak to actually support a key role for this TF in N and Septoria responses. As previously commented, WRKY is a very large family, and many members are associated to TFs and signal regulators potentially related to both abiotic and biotic stresses. In fact, other WRKY factors interacting broad range stress response regulators can be detected by searches in the STRING database using your parameters.